# Immune Resistance in Lung Adenocarcinoma

**DOI:** 10.3390/cancers13030384

**Published:** 2021-01-21

**Authors:** Magda Spella, Georgios T. Stathopoulos

**Affiliations:** 1Laboratory for Molecular Respiratory Carcinogenesis, Department of Physiology, Faculty of Medicine, University of Patras, Rio, 26504 Achaia, Greece; magsp@upatras.gr; 2Comprehensive Pneumology Center (CPC), Institute for Lung Biology and Disease (iLBD), Helmholtz Center Munich–German Research Center for Environmental Health, Member of the German Center for Lung Research, 81377 Munich, Germany

**Keywords:** non-small cell lung cancer, lung adenocarcinoma, immune resistance, targeted therapies, immunotherapy

## Abstract

**Simple Summary:**

Lung adenocarcinoma (LUAD) is the major subtype of lung cancer and represents the deadliest human cancer, affecting current-, ex-, and even non-smokers. LUAD is driven by the accumulation of mutations in several different genes, which results in uncontrolled proliferation of the lung cells and the formation of tumors. Our immune system can recognize these transformed cancer cells and start an immune response in order to eliminate them. Unfortunately, cancer cells in turn adapt to and ultimately evade host immune defenses, fostering their growth. Current therapeutic approaches for cancer patients called “immunotherapies” aim to overcome the cancer’s ability to develop mechanisms of resistance to our immune defenses. This review summarizes the mechanisms used by LUAD cancer cells to develop immune resistance and discusses current and future therapeutic approaches in LUAD patient management.

**Abstract:**

Lung cancer is the leading cancer killer worldwide, imposing grievous challenges for patients and clinicians. The incidence of lung adenocarcinoma (LUAD), the main histologic subtype of lung cancer, is still increasing in current-, ex-, and even non-smokers, whereas its five-year survival rate is approximately 15% as the vast majority of patients usually present with advanced disease at the time of diagnosis. The generation of novel drugs targeting key disease driver mutations has created optimism for the treatment of LUAD, but, as these mutations are not universal, this therapeutic line benefits only a subset of patients. More recently, the advent of targeted immunotherapies and their documented clinical efficacy in many different cancers, including LUAD, have started to change cancer management. Immunotherapies have been developed in order to overcome the cancer’s ability to develop mechanisms of immune resistance, i.e., to adapt to and evade the host inflammatory and immune responses. Identifying a cancer’s immune resistance mechanisms will likely advance the development of personalized immunotherapies. This review examines the key pathways of immune resistance at play in LUAD and explores therapeutic strategies which can unleash potent antitumor immune responses and significantly improve therapeutic efficacy, quality of life, and survival in LUAD.

## 1. Introduction

Lung cancer represents one of the most challenging health problems, claiming each year more lives than breast, prostate and pancreatic cancer combined [1]. Non-small-cell lung cancer (NSCLC) accounts for approximately 85% of lung cancer incidence, and lung adenocarcinoma (LUAD), NSCLC’s most prevalent subtype, is still increasing in current-, ex-, and even non-smokers [2]. Patients with localized and early stage disease receive standard surgery, but the vast majority of patients are usually diagnosed with advanced disease, receive conventional therapies such as combination chemotherapy and radiation, and face a high mortality rate.

In the classical sense, cancer is driven by genetic and epigenetic aberrations and the complicated cross-talk between many different pathways, involving mainly mutations activating proto-oncogenes and/or suppressing tumor suppressors and resulting in uncontrolled cell proliferation. Targeted therapies have been developed for some of these genetic aberrations, rendering the molecular characterization of a patient’s tumor an imperative prerequisite for the successful design of the most efficient targeted treatment strategy. Among the most commonly found mutations in LUAD, caused spontaneously, by genetic predisposition, or by cigarette smoking and other noxious inhaled agents, are activating mutations in epidermal growth factor receptor (*EGFR*, 14%), Kirsten rat sarcoma virus (*KRAS*, 33%), mesenchymal epithelial transition factor proto-oncogene (*MET*, 7%), B-Raf proto-oncogene (*BRAF*, 10%), and Phosphatidylinositol-4,5-Bisphosphate 3-Kinase Catalytic Subunit Alpha (*PIK3CA*, 7%), mutations in tumor suppressor genes like tumor protein p53 (*TP53*, 46%) and serine/threonine kinase 11 (*STK11*, 17%), and translocations in anaplastic lymphoma kinase (*ALK*, 3–7%), ROS proto-oncogene 1(*ROS1*, 2%), or Ret proto-oncogene (*RET*, 1–2%) [3]. In the case of *EGFR* activating mutations, first-line treatment includes EGFR tyrosine kinase inhibitors (EGFR-TKI, like gefitinib, erlotinib, afatinib, and osimertinib), whereas *BRAF*-mutant tumors can be treated with dabrafenib and trametinib [4,5,6]. Encouraging results have also been demonstrated for patients harboring *ALK* (crizotinib, ceritinib, alectinib, brigatinib, and lorlatinib), *ROS1* (crizotinib, ceritinib, and lorlatinib) or *RET* translocations (cabozantinib) [5,6,7,8]. Crizotinib can be also effective against *MET*-mutant cancers [5]. The inhibitors of PIK3CA signaling pathway, temsirolimus and everolimus, are currently under clinical trials [6]. Unfortunately, these treatments benefit only a small subset of LUAD patients, not only because all other mutated genes are not yet effectively clinically targeted, but also due to the development of primary and secondary resistance [9]. *KRAS*, the most commonly mutated driver oncogene in LUAD, remains notoriously untargeted. What is more, its mutations are mutually exclusive with *EGFR* mutations, so patients with *KRAS* mutations are resistant to *EGFR* targeted treatment and are faced with no efficient treatment options and a poor prognosis [10,11]. Encouragingly, the KRAS^G12C^ inhibitor sotorasib can potentially be the first approved targeted therapy for patients with *KRAS^G12C^*-mutant NSCLC [12].

The emergence of deregulated host inflammatory pathways as an important hallmark of cancer development [13] has added great complexity to the above classical sense of cancer pathogenesis, but has also provided an alternative approach for novel therapeutic strategies. Chronic lung inflammation associated with tobacco smoking is strongly implicated in LUAD development, highlighting the dynamic communication of lung tumors with the surrounding tumor microenvironment (TME), consisting of the interface of the bronchoalveolar compartment with host immune cells, cytokines, chemokines, and other components [13]. Developing tumors can hijack and evade host immune surveillance mechanisms [13], and immunotherapies aim to stimulate the patient’s immune system to elicit a competent anti-tumor immune response. Several such approaches have been developed, including vaccine therapy [14], chimeric antigen receptor (CAR) T cells [15], and immune checkpoint inhibitors (ICI) like antibodies against cytotoxic T-lymphocyte-associated antigen 4 (CTLA-4) [16], programmed cell death 1 (PD-1) and programmed cell death ligand 1 (PD-L1) [17,18,19], and are currently used in clinical practice for the treatment of several cancers, including lung cancer [20]. Despite these encouraging results, increased immune tolerance is often documented in many cancers [21], implying that there is more to learn about the cross-talk between the developing tumors and the immune cells within the TME. Interestingly, *KRAS* mutations are increasingly shown to affect tumor interactions with the surrounding TME [11,22,23,24,25]. Below we will discuss the main mechanisms of tumor immune resistance, with a special emphasis on LUAD and mutant *KRAS*-mediated effects underlying immune resistance mechanisms.

## 2. Tobacco Smoke Immunomodulatory Effects

LUAD development is strongly associated with environmental exposures like cigarette smoking. Tobacco smoke contains many chemical constituents and irritant particles which are deposited on the airways and cause repeated or chronic pulmonary inflammation, sculpturing the TME [26]. The inflammatory response is initiated with the influx of neutrophils and macrophages, resulting in the secretion of signaling molecules like tumor necrosis factor (TNF) and interleukins (IL) IL-6, IL-8, and IL-1β [27,28], and in the activation of signaling cascades involving nuclear factor-κB (NF-κB), mutant RAS or ERK [29,30,31]. This ultimately leads to the secretion of immunosuppressive factors, permitting the malignant transformation of lung epithelial cells [27,30,32]. Interestingly, lung tumors of smokers and non-smokers display very distinct inflammatory signatures, with marked differences in mast cell and CD4+ T cell numbers [33]. Furthermore, it has been postulated that activated neutrophils can enhance the genomic instability caused by tobacco smoke carcinogens in lung epithelial cells, suggesting that inflammation can actually act as a tumor initiator or promoter [27,34].

## 3. Tumor Heterogeneity

LUAD tumors are characterized by great heterogeneity, as evidenced by the thousands of different mutations identified per cancer genome, including single nucleotide variants (SNV), copy number alterations (CNA), gene fusions, and major chromosomal events [3,35]. The differences between smokers and non-smokers in the TME described in the previous paragraph are also mirrored in the different genomic LUAD profiles between the two groups, with smokers displaying higher mutations burdens [3]. Of course, mutational hotspots are also found in LUAD, involving activating mutations in *EGFR*, *KRAS*, *BRAF*, and *PIK3CA*, deleterious mutations in tumor suppressor genes like *TP53* and *STK11*, and balanced inversions (translocations or fusions) in *ALK* or *RET* [3]. A current mutation plot and mutual exclusivity and co-occurrence information of frequent cancer genes from the cBioportal analyzing the TCGA (The Cancer Genome Atlas) pan-cancer LUAD dataset are shown in Figure 1 and Table 1, respectively. Indicative of the bewildering tumor heterogeneity evident in the plot is also the fact that even among patients with the same smoking status or the same driver mutation, a marked intratumoral heterogeneity will be observed, as each tumor contains a plurality of populations of cancer cells, with different genetic and epigenetic features [36]. Intratumoral heterogeneity also assumes that different cell lineages can be targeted by drug treatment, as some subpopulations, containing the suitable genetic disposition, will display resistance. Under the constant pressure within the TME, tumor cells, exhibiting a remarkable plasticity, modulate their antigenic landscape, evade immune surveillance, survive and continue to propagate. These cells can then drive expression of signaling effectors which will favor tumor growth. Intratumoral heterogeneity remains a constant threat to immunotherapies.

## 4. Inflammatory Interactions of LUAD Cells with the Tumor Microenvironment Mediating Immune Escape

LUAD tumor cells establish a complex surrounding TME rich in inflammatory signaling, orchestrated by expression of cytokines, chemokines, and their cognate receptors by both cancer and host cells [37]. This inflammatory signaling results in the attraction of various immune cellular populations, including tumor-associated macrophages (TAMs), tumor-reactive lymphocytes, myeloid-derived suppressor cells (MDSCs), tumor-associated neutrophils and mast cells, which interact with tumor cells to ultimately shape a highly immune suppressive TME, with diminished tumor cytotoxic and enhanced tumor-promoting manifestations [38,39,40].

TAMs are the most crucial immune cell type of the TME, favor cancer progression, and are associated with poor prognosis [39,41]. Macrophages are generally classified into two distinct and reactive populations, perpetually changing their phenotype as they dynamically respond to environmental alterations of the TME. M1 or classically activated pro-inflammatory macrophages are typically activated by IFN-γ and lipopolysaccharide, express IL-12 and mediate anti-tumor responses. On the other hand, M2 or alternatively activated macrophages are typically activated by IL-4, IL-13 and IL-10, express IL-10 and favor wound healing as well as tumor development by suppressing anti-tumor T cell responses [40].

Pulmonary alveolar macrophages have also been implicated in lung cancer pathobiology. Resident alveolar macrophages display significant immunomodulatory properties and an M2 phenotype in the naïve state. They adopt immunosuppressive roles in preventing unwanted immune responses to environmental antigens that bombard the lungs every day. It has been postulated that these immunosuppressive functions can be hijacked by tumors to their advantage, and although the exact role of alveolar macrophages in the pathobiology of lung cancer remains controversial, recent evidence suggests that they can be preconditioned by primary tumors to suppress immune responses in the lungs, thereby facilitating cancer progression [42].

MDSCs are immune suppressive myeloid cells fostering tumor progression in many different ways, most of which result in the inhibition of activation of tumor-reactive T cells and of natural killer (NK) cell cytotoxicity [40]. Evidence supports a key role of the spleen as an intermediate organ in their phenotypic education, prior to their migration from the bone marrow to tumor sites [40] Figure 1 and Table 1.

Tumor infiltrating T cells are key players in the establishment of a favorable TME. Among them, T regulatory cells (Treg) exhibit immune suppressing functions, which, through the secretion of IL-10 and transforming growth factor (TGF)-β, help cancer cells evade the immune defensive mechanisms [45]. Lung tumor cells express immunosuppressive factors, such as IL-10 and TGFβ, contributing to the recruitment Tregs and MDSCs [46,47].

*KRAS* mutations are common activating mutations in LUAD (Figure 1). *KRAS*-mutant LUAD appear to be densely infiltrated by myeloid-derived immune cells (mainly macrophages, neutrophils and eosinophils) [48], indicating that there might be a correlation between *KRAS* mutations and immune cell specificity within the TME. Experimental models of LUAD formation provide a unique tool for the dissection of the mechanisms adopted by tumor cells to recruit inflammatory cells to the TME and to thereafter hijack the host immune repertoire to their advantage. By using genetically modified mice permitting the oncogenic overexpression of mutant *Kras* in bronchial club epithelial cells expressing Clara cell secretory protein (CC10), Ji et al. showed an overwhelming neutrophil attraction to the TME accompanying LUAD formation and identified mutant *Kras* tumor-elaborated CXC chemokines (and their human orthologues IL-8 and CXCL-5) as the crucial mediators for this mechanism [49]. Mutant *KRAS*-mediated expansion of myeloid-derived suppressor cells resulted in the suppression of the antitumor activity of CD8+ cytotoxic T cells [50].

Cross-talk within the TME between tumor and host immune cells resulting in pro-tumorigenic effects is increasingly elucidated in the context of other thoracic malignancies as well. For LUAD tumor cells with the ability to metastasize to the pleura and foster a malignant pleural effusion (MPE), it has been shown that the determining factor is the presence of *KRAS* activating mutations, which favor the production of CCL2 by the tumor cells. In turn, tumor-elaborated CCL2 paracrine signaling attracts CD11b+GR1+ myeloid cells to the pleural cavity [23]. This axis was further elucidated by the identification of an additional important tumor-host inflammatory circuit at play in MPE formation, which provided important insights on how an oncogene can co-opt the TME to favor tumor progression. Specifically, mutant *KRAS* tumor cells were shown to respond to IL-1β secretion from the CCL2-recruited myeloid cells by activating the alternative signaling cascade of NF-κB pathway, forming in this way a circuit which fuels the secretion of CXCL1 by tumor cells, enhances tumor and MPE development and drives drug resistance [25]. Additionally, LUAD tumor cells used in a model of pleural carcinomatosis were found to secrete the chemoattractant CCL2 and the protein osteopontin (SPP1) in order to mediate mast cells recruitment and degranulation [51]. In the same study, the adenocarcinoma cells were also shown to trigger myeloid cells to secrete IL-1β [51]. Mast cell-derived IL-1β was also documented as a factor fostering LUAD development in a study which utilized experimental mouse models of carcinogen-induced LUAD to show the significance of mast cells in LUAD progression [24].

IL-1β is an inflammatory mediator implicated in many different malignancies. In NSCLC IL-1β has been evaluated as a prognostic marker portending poor survival [52] and the above observations clearly establish a link between IL-1β and *KRAS* mutant lung cancer. Tobacco smoke also causes the bronchial epithelium to produce IL-1β [53]. This link of IL-1β to lung cancer has been recently clinically tested, with fascinating results. In a clinical trial assessing the anti-inflammatory effects on cardiovascular disease of a novel inhibitor of IL-1β (Canakinumab, ACZ885, Ilaris), the Canakinumab Anti-Inflammatory Thrombosis Outcomes Study (CANTOS), the authors also documented an impressive decrease in NSCLC incidence and mortality, an outcome recognizing the potential of IL-1β inhibition in NSCLC [54,55]. At the molecular level, mitigating IL-1β signaling would prevent the oncogenic activation of the alternative pathway of NF-κB signaling in lung cancer cells, impairing in this way the cascade initiated by *KRAS* mutant tumor cells [25]. Enhanced IL-1β secretion by tumor-associated neutrophils has also been described as a mechanism mediating drug resistance in experimental models of lung cancer [56], raising the hope that the inhibition of IL-1β signaling could also diminish the occurrence of tumor resistance incidences.

*KRAS*-mutant tumors are characterized by large amount of neoantigens [22], and as such they are faced with the need to tackle the immune response mediated by tumor-specific T cells in order to sustain their growth. Along this line, it has been documented that *KRAS* activation downregulates expression of MHC I at the cell surface, thus impairing recognition by CD8+ T cells and promoting immune evasion [57,58]. Furthermore, *KRAS* mutations upregulated expression of the suppressive cytokines IL-10 and TGFβ1, which converted CD4+ T helper cells into T regulatory cells (Tregs) [59]. Tregs are immunosuppressive cells which have been identified to play important roles during *KRAS* mutant tumorigenesis [60]. In the lung epithelium, activating *KRAS* mutations resulted in elevated numbers of Th17 cells, which in turn produced high levels of IL17. This cytokine then activated a cascade involving the acceleration of cell proliferation, the production of metalloproteases (MMP-7 and MMP-12) and of additional proinflammatory cytokines like IL6, CCL2, and CXCL2, and ultimately the attraction of MDSCs to the lung [61,62]. Interestingly, differences have been documented between *KRAS* mutant and *EGFR* mutant LUAD relating to their immune content, with the former presenting increased populations of Tregs, IL17-producing lymphocytes and reduced NK cells [48]. Finally, mutant *KRAS* silencing led to increased expression of MHC I surface proteins on tumor cells and production of IL-18, inducing ultimately tumor regression [63].

The above observations are of particular clinical interest if *KRAS* activating mutations affect the response or the resistance to currently licensed immunotherapies. After all, it is well documented that immune checkpoint blockade treatment has durable beneficial effects for patients with NSCLC [12,64,65,66,67], and even more so for lung cancer patients harboring mutations in both *TP53* and *KRAS* [68]. Supporting this notion, PD-L1 overexpression has been found to be mediated through the mutant *KRAS*-ERK signaling axis [69,70], and the highest level of PD-L1 expression was found in *KRAS/TP53* co-mutated lung tumors [68]. Accumulating evidence suggest that *KRAS* mutations that coexist with wild-type *STK11* co-operate for establishment of an immunosuppressive TME [71,72,73].

It is now widely accepted that KRAS extensively regulates the cross-talk between cancer and host immune cells, promoting the switch from an anti-tumor to a pro-tumor response and the development of immune escape mechanisms.

## 5. Adaptive Immune Resistance

Tumors are rich sources of neoantigens, formed as a sequence of their somatic mutations, which are foreign to the immune system and therefore capable of eliciting a robust antitumor immune response [74]. This indicates that tumors can progressively foster their growth by adopting mechanisms permitting tumor neoantigens to go unnoticed by the host immune defenses. The term adaptive immune resistance describes the process in which tumor cells change their phenotype, in a dynamic and reactive fashion, in order to avoid a host immune attack orchestrated by tumor neoantigen-specific cytotoxic T cells [75]. The adaptive immune resistance is the product of the cross-talk between tumor and host immune cells within the TME. Normally, T cell activation requires the interaction of antigen-specific T cells with the cognate tumor antigen. After T cell activation, signaling through the T-cell receptor (TCR) induces a robust immune response, triggered by secretion of interferon, and the expression of regulatory receptors (such as PD-1 and CTLA-4), which will act as a negative feedback mechanism. For this to happen, interferon also induces the expression of PD-L1, and the molecular interaction between the receptor and the ligand limits the inflammatory response [19]. This adaptive process is used by the tumor, which expresses PD-L1 and thereby turns off PD-1-positive T cells [76]. Usually PD-L1 expression is restricted at the invasive margin of the tumor, an area with abundant T cells, suggesting that PD-L1 expression is a counteracting mechanism adopted by tumor cells as a consequence of the presence of tumor antigen-specific T cells [77,78]. Interestingly, PD-L1 expression is not only triggered in cancer cells but also on the surface of dendritic cells, macrophages, myeloid-derived cells, TME stromal cells, and even tumor infiltrating T cells [77,79,80,81]. Another alternative is that PD-L1 overexpression is the result of gene amplification events of chromosome 9 [82]. Disruption of this negative feedback using anti-PD-1, anti-PD-L1, or anti-CTLA4 antibodies reactivates T-cell cytotoxic properties which effectively lead to tumor antigen-specific immune responses and tumor killing [16,17,18,19].

The production of proinflammatory cytokines within the TME may also result in tumor adaptive changes and immune evasion. It has been shown that tumor necrosis factor-a (TNF-α) production by infiltrating T cells promoted a dedifferentiating process in melanoma cancer cells, thereby leading to tumor antigen loss and an adaptive immune escape mechanism [83]. This process resembles the epithelial-to-mesenchymal transition (EMT), and in fact many aspects of EMT are related to inflammatory cytokines like IL6, TNFα and tumor growth factor-β TGFβ [83,84,85]. In this way, an adaptive immune resistance mechanism can even be a tumor promoting mechanism.

## 6. Acquired Immune Resistance

Blocking the PD-1 and CTL-4 axes with specific antibodies (like the anti-PD-1 molecules nivolumab, pembrolizumab and the emerging cemiplimab, the anti-PD-L1 antibodies atezolizumab and durvalumab, and the anti-CTL-4 antibody ipilimumab) exhibits durable beneficial effects for patients with NSCLC, with or without *KRAS* mutations [64,65,66,67,86,87].

Unfortunately, not all patients with LUAD can benefit from blockade of immune checkpoints, and many of the initial responders will eventually develop resistance to the therapy [88], while the mechanisms behind this phenomenon are currently under intense scrutiny.

For patients who are refractory to immune checkpoint blockade, one possible explanation is that their tumors contain low mutational load and they lack immunogenic tumor antigens, such as nonsmoker LUAD [89,90]. In conjunction with low mutation burden, the activation of MAPK in *EGFR*-mutant LUAD contributed in the establishment of highly immunosuppressive conditions due to increased recruitment of Tregs and TAMs [91].

For patients with acquired resistance to immunotherapies, possible explanations are the upregulation of alternative immune checkpoints, mainly T-cell immunoglobulin mucin-3 (TIM-3) [92], loss of HLA haplotypes due to rearrangement of chromosome 6 [93,94,95] and somatic mutations in interferon receptor-associated Janus kinase 1 (*JAK1*) or Janus kinase 2 (*JAK2*) and antigen-presenting protein beta-2-microglobulin [*B2M*, leading to loss of surface expression of major histocompatibility complex class I (MHC I)] [96,97]. Loss of tumor neoantigens through elimination of tumor subclones or chromosomal deletions has also been described as a mechanism of immune edited acquired resistance in lung cancer, in which tumors responded to the selective pressure of immune checkpoint blockade by eliminating mutations affecting MHC and TCR binding [98]. Similarly, immunoediting induced by immune checkpoint blockade transformed NSCLC to SCLC (small cell lung cancer) by selectively eliminating the treatment-sensitive tumor cells [99,100].

## 7. Drug Resistance

Cancer cells can become excessively dependent on specific driver mutations, a characteristic defined as “oncogene addiction” [101]. As a result, many targeted therapies have been developed aiming to inhibit these oncogenic driver mutations and genotype-directed therapy in advanced NSCLC has led to significant improvements in overall survival [102]. For patients with *EGFR* somatic mutations, administration of small molecule TKIs of EGFR (gefitinib, erlotinib, afatinib, and osimertinib) is the standard course of treatment, resulting in high response rates and prolonged progression-free survival [4]. Encouraging results have also been demonstrated for patients harboring *ALK* rearrangements, after receiving treatment with ALK-TKIs (crizotinib, ceritinib, and alectinib) [7]. The effectiveness of TKI therapy is however limited by the ability of cancer cells to evolve under the pressure of the therapy and to ultimately acquire resistance [9].

A total of 4–10% of newly diagnosed NSCLC patients will not respond to TKI therapy, exhibiting what is called primary (intrinsic) resistance [9]. One mechanism underlying intrinsic resistance could be the existence of activating yet non-sensitizing driver mutations, as has been described for the *EGFR*-T790M mutation with reported frequencies varying from <10% to 65% [103,104]. The combination of *EGFR* mutations with other genetic alterations has also been shown to contribute to intrinsic resistance to TKIs, like *MET* amplification and *BCL2L11* mutation [104,105]. Another potential modulator conferring intrinsic resistance to EGFR-TKIs is NF-κB. It has been reported that knocking down NF-κB in *EGFR*-mutant lung cancer cells enhanced erlotinib sensitivity, while patient response to erlotinib was associated with high expression of the NF-κB inhibitor IκBα [106], raising hope that a combinatorial treatment targeting both EGFR and NF-κB can have beneficial responses in the clinic.

Unfortunately, even patients who initially respond positively to TKIs administration, usually develop secondary or acquired resistance to the treatment followed by disease progression. Most commonly, this is attributed to secondary somatic mutations of the target gene which hinder the binding of the drug and confer in this way resistance to treatment. In *EGFR*-mutant NSCLC, the dominant cause of TKI acquired resistance is the *EGFR*-T790M mutation, accounting for more than half of the incidences of reported resistance [105,107]. The most common secondary mutations found in *ALK*-mutant NSCLC are the *ALK*-L1196M and *ALK*-G1269A mutations, causing resistance by interfering with TKI binding [108,109]. Another mechanism responsible for acquired resistance to TKI therapy is gene target amplification. This has been documented with both mutant *EGFR* and *ALK*, and gene amplification can occur alone or in combination with the secondary somatic resistance mutations [107,109]. LUAD can also adapt to the pressure of TKIs by activating bypass signaling pathways, which allow cancer cells to continue to grow despite the targeting of the driver gene mutations, like PI3K/AKT or RAF/MEK/ERK pathways [3,110], or by triggered mutations activating downstream effector molecules of the targeted cascade, like MAPK activation in both *EGFR-* and *ALK*-mutant NSCLC [111,112]. NSCLC has also been shown to phenotypically transform to SCLC in order to resist TKIs treatment, resulting in the formation of an *EGFR-* or *ALK*-mutant SCLC, which is insensitive to TKIs therapy [107,113]. Finally, the notion of “drug tolerant cells” has been also suggested to explain both intrinsic and acquired resistance, which dictates that small populations of cancer cells can tolerate the drug exposure, possibly due to epigenetic changes, persist under a quiescent state and propagate until a more permanent resistance mechanism is acquired [114].

Conceptually, all resistance mechanisms can be viewed as the dynamic manifestations of cancer evolution in order to overcome the selective pressure of targeted therapies.

## 8. Strategies to Overcome Resistance

As cancer resistance mechanisms are under intense research scrutiny, the culprits of these endeavors have shed some light on the profound complexities of tumor biology and have guided the design of new therapeutic strategies. Immune checkpoint blockers are currently at the forefront of cancer therapeutics, with many clinical trials underway assessing the efficacy of the treatment to many solid cancers, including LUAD [115]. Significant research has also been done towards the use of CAR T cells in the very challenging context of solid tumors [15]. Despite the obstacles, the preclinical combination of CAR T cells with checkpoint immune inhibitors has demonstrated encouraging results and is now under clinical trials for *EGFR* mutant tumors [116]. Furthermore, strategies targeting Tregs, TAMs and MDSCs are also being evaluated [47,117], along with the development of newer generation TKIs and combinatorial drug treatments to overcome resistance [9].

Accumulating evidence supports the pivotal role of KRAS not only in governing cancer cells’ autonomous mechanisms of proliferation, but also in controlling the immune landscape within the TME and in mediating immune escape and ultimately tumor progression. Although the gene remains largely untargeted, pioneering studies have elucidated the signaling cascades originating in lung cancer cells from *KRAS* activating mutations and have provided the medical community with novel therapeutic targets, such as the combined *KRAS*/NF-κB inhibition [25]. Of note, inhibition of IL-1β, another downstream effector of mutant *KRAS*, is currently evaluated for NSCLC with very promising results [54,55].

## 9. Conclusions

Immunotherapies hold great promise in extending the survival of patients with LUAD. Despite these significant advances, cancers still find ways to counteract any therapeutic strategy, by dynamically evolving and developing resistance mechanisms. Further and deeper understanding of the cross-talk between cancer and host immune cells within the TME will broaden our knowledge on the fundamental mechanisms of tumor escape from the immune system and will hopefully advance precision medicine and personalized combination immunotherapy. For example, the significance of the frequent co-alterations in immune genes observed in LUAD (such as the genes mentioned in this text, analyzed in Figure 2 and Table 2), have received far less attention than cancer drivers, and need to be functionally explored in the future in terms of their cellular localization and function. In conclusion, LUAD is probably one of the best studied bodily cancers and immunotherapy against this form of lung tumors has come a long way. Tackling immune resistance of this dreadful tumor will be the next great challenge for thoracic oncology.

## Figures and Tables

**Figure 1 cancers-13-00384-f001:**
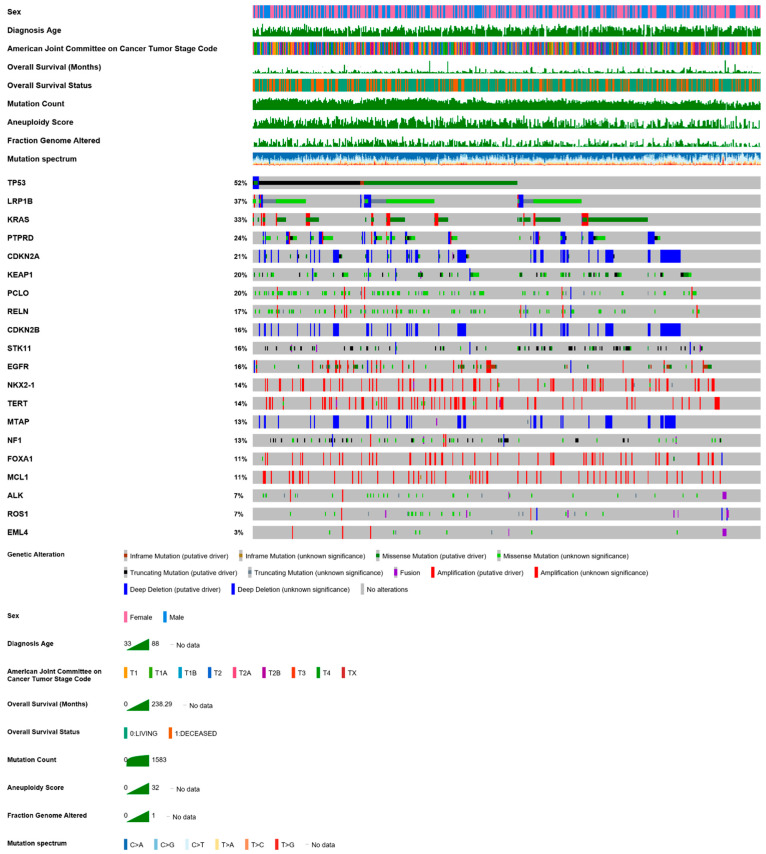
A current mutation plot of the most frequently altered genes in the TCGA pan-cancer Lung adenocarcinoma (LUAD) dataset (*n* = 507). Queried genes (*TP53*, *LRP1B*, *KRAS*, *PTPRD*, *CDKN2A*, *KEAP1*, *PCLO*, *RELN*, *CDKN2B*, *STK11*, *EGFR*, *NKX2-1*, *TERT*, *MTAP*, *NF1*, *FOXA1*, *MCL1*, *ALK*, *ROS1*, *EML4*) were altered in 475 (94%) of patients. Note the heterogeneity and mosaic complexity of mutations in key cancer genes. Note also the multiple mutual exclusivity and addiction patterns that are used to provide clues to which genes are candidate drivers or passengers. Data from https://www.cbioportal.org/study/summary?id=luad_tcga_pan_can_atlas_2018 [43,44].

**Figure 2 cancers-13-00384-f002:**
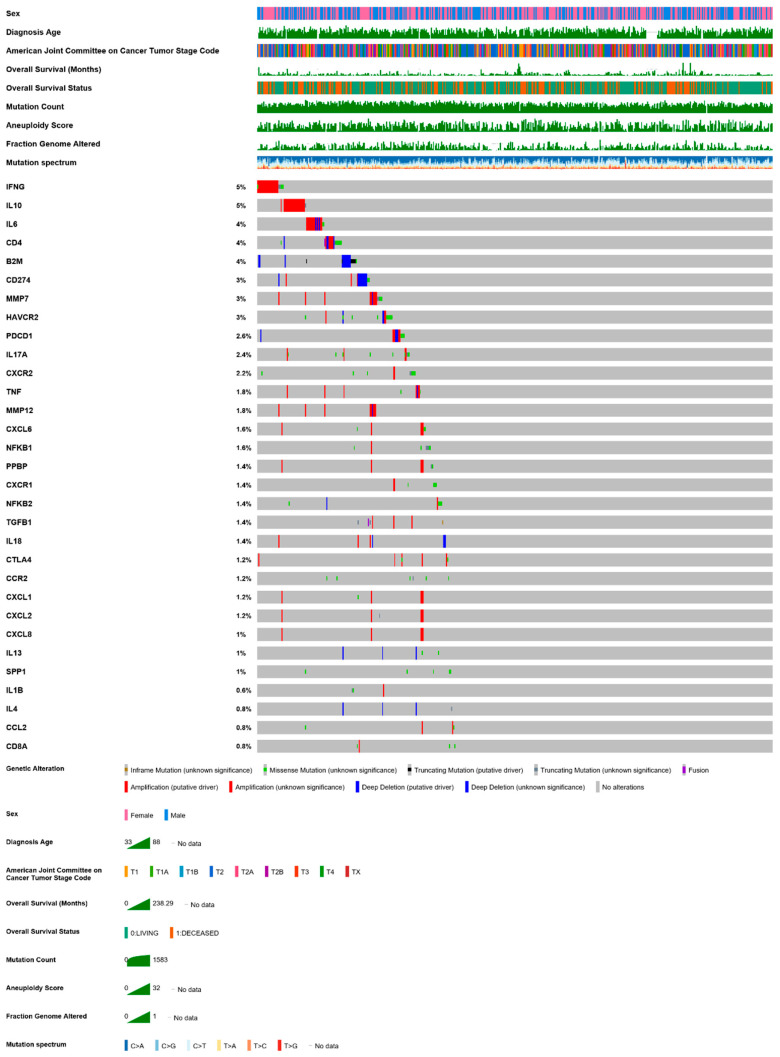
A mutation plot of altered immune genes reviewed in this text in the TCGA pan-cancer LUAD dataset (*n* = 507). Queried genes (*IFNG*, *IL10*, *IL6*, *CD4*, *B2M*, *CD274*, *MMP7*, *HAVCR2*, *PDCD1*, *IL17A*, *CXCR2*, *TNF*, *MMP12*, *CXCL6*, *NFKB1*, *PPBP*, *CXCR1*, *NFKB2*, *TGFB1*, *IL18*, *CTLA4*, *CCR2*, *CXCL1*, *CXCL2*, *CXCL8*, *IL13*, *SPP1*, *IL1B*, *IL4*, *CCL2*, *CD8A*) were altered in 135 patients (98%). Data from https://www.cbioportal.org/study/summary?id=luad_tcga_pan_can_atlas_2018 [43,44].

**Table 1 cancers-13-00384-t001:** Mutual exclusivity and co-occurrence of frequently altered cancer genes in the TCGA pan-cancer LUAD dataset (*n* = 507). Queried genes were frequently mutually exclusive (MUEX; indicating dominant oncogenes) or co-occurring (COOC; indicating oncogene addiction). *p*, probability, hypergeometric test; *q*, probability, false discovery rate. Data are from https://www.cbioportal.org/study/summary?id=luad_tcga_pan_can_atlas_2018 [43,44].

A	B	Neither	A Not B	B Not A	Both	Log2 Odds Ratio	*p*-Value	*q*-Value	Tendency
CDKN2A	CDKN2B	402	22	0	83	>3	<0.001	<0.001	COOC
CDKN2B	MTAP	422	21	2	62	>3	<0.001	<0.001	COOC
CDKN2A	MTAP	400	43	2	62	>3	<0.001	<0.001	COOC
NKX2-1	FOXA1	433	20	5	49	>3	<0.001	<0.001	COOC
PCLO	RELN	361	58	45	43	2.572	<0.001	<0.001	COOC
KRAS	EGFR	265	163	73	6	−2.904	<0.001	<0.001	MUEX
LRP1B	RELN	288	131	33	55	1.873	<0.001	<0.001	COOC
LRP1B	PCLO	278	128	43	58	1.551	<0.001	<0.001	COOC
TP53	LRP1B	179	142	64	122	1.265	<0.001	<0.001	COOC
TP53	PCLO	215	191	28	73	1.553	<0.001	<0.001	COOC
ALK	EML4	462	28	9	8	>3	<0.001	<0.001	COOC
KEAP1	STK11	359	68	49	31	1.74	<0.001	<0.001	COOC
TP53	KRAS	139	199	104	65	−1.196	<0.001	<0.001	MUEX
KRAS	STK11	301	126	37	43	1.473	<0.001	<0.001	COOC
TP53	TERT	225	212	18	52	1.616	<0.001	<0.001	COOC
TP53	NF1	226	216	17	48	1.563	<0.001	0.001	COOC
LRP1B	PTPRD	261	124	60	62	1.121	<0.001	0.002	COOC
CDKN2A	ROS1	384	89	18	16	1.939	<0.001	0.003	COOC
TP53	EGFR	219	209	24	55	1.264	<0.001	0.005	COOC
KRAS	NF1	283	159	55	10	−1.628	<0.001	0.005	MUEX
LRP1B	EGFR	258	170	63	16	−1.375	<0.001	0.005	MUEX
TP53	RELN	215	204	28	60	1.175	<0.001	0.005	COOC
PTPRD	PCLO	321	85	64	37	1.126	0.001	0.009	COOC
STK11	TERT	360	77	67	3	−2.256	0.002	0.013	MUEX
PTPRD	RELN	329	90	56	32	1.063	0.003	0.022	COOC
KRAS	ROS1	308	165	30	4	−2.006	0.003	0.022	MUEX
KEAP1	EGFR	336	92	72	7	−1.494	0.005	0.033	MUEX
STK11	EGFR	353	75	74	5	−1.653	0.006	0.041	MUEX
TP53	PTPRD	197	188	46	76	0.792	0.006	0.041	COOC
TP53	STK11	194	233	49	31	−0.925	0.007	0.042	MUEX
LRP1B	KRAS	227	111	94	75	0.706	0.007	0.046	COOC

**Table 2 cancers-13-00384-t002:** Co-occurrence of frequently altered immune genes in the TCGA pan-cancer LUAD dataset (*n* = 507). Queried genes (*IFNG*, *IL10*, *IL6*, *CD4*, *B2M*, *CD274*, *MMP7*, *HAVCR2*, *PDCD1*, *IL17A*, *CXCR2*, *TNF*, *MMP12*, *CXCL6*, *NFKB1*, *PPBP*, *CXCR1*, *NFKB2*, *TGFB1*, *IL18*, *CTLA4*, *CCR2*, *CXCL1*, *CXCL2*, *CXCL8*, *IL13*, *SPP1*, *IL1B*, *IL4*, *CCL2*, *CD8A*) were never mutually exclusive but frequently co-occurring (COOC; indicating oncogene addiction). *p*, probability, hypergeometric test; *q*, probability, false discovery rate. Data are freely available at https://www.cbioportal.org/study/summary?id=luad_tcga_pan_can_atlas_2018 [43,44].

A	B	Neither	A Not B	B Not A	Both	Log2 Odds Ratio	*p*-Value	*q*-Value	Tendency
MMP7	MMP12	492	6	0	9	>3	<0.001	<0.001	Co-occurrence
CXCL1	CXCL8	501	1	0	5	>3	<0.001	<0.001	Co-occurrence
CXCL2	CXCL8	501	1	0	5	>3	<0.001	<0.001	Co-occurrence
PPBP	CXCL8	500	2	0	5	>3	<0.001	<0.001	Co-occurrence
CXCL1	CXCL2	500	1	1	5	>3	<0.001	<0.001	Co-occurrence
CXCL6	CXCL8	499	3	0	5	>3	<0.001	<0.001	Co-occurrence
PPBP	CXCL1	499	2	1	5	>3	<0.001	<0.001	Co-occurrence
PPBP	CXCL2	499	2	1	5	>3	<0.001	<0.001	Co-occurrence
CXCL6	CXCL1	498	3	1	5	>3	<0.001	<0.001	Co-occurrence
CXCL6	CXCL2	498	3	1	5	>3	<0.001	<0.001	Co-occurrence
CXCL6	PPBP	497	3	2	5	>3	<0.001	<0.001	Co-occurrence
IL13	IL4	501	2	1	3	>3	<0.001	<0.001	Co-occurrence
TGFB1	IL18	496	4	4	3	>3	<0.001	0.002	Co-occurrence
MMP12	IL18	494	6	4	3	>3	<0.001	0.004	Co-occurrence
MMP7	IL18	488	12	4	3	>3	<0.001	0.021	Co-occurrence

## Data Availability

Data sharing not applicable.

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
