# Peer review of "Immune Resistance in Lung Adenocarcinoma"

_cancers, 2021, doi:10.3390/cancers13030384_

Round 1

Reviewer 1 Report

Minor points:

  • Line 163:  What is meant by "pinned" mutant Kras?
  • Line 232: is there a word missing after "of foreign"?
  • Around line 249: Perhaps important to cite / mention work studying dendritic cell PDL1 in Mayoux(Chen) 2020 and Soyong(Mellman) 2020 ?
  • Line 263: Why no mention of Imfinzi or Libtayo?

Major points:

  • Table2:  The genes examined are limited by those in current common panels.  Future reviews will hopefully include WES and/or a focus on IO-related genes. e.g. HLA/ Agpres
  • Lines 278-284:  why no mention of a role HLA by McGranahan/2017, Chowell/2019, Montesion 2020.  These studies cannot be omitted.

Point of agreement:

  • Line 356:  This reviewer fully agrees with lack of relative emphasis on frequent co-mutations vis a vis driver mutations.  In fact, there is little study of "immunologic drivers".

Author Response

RESPONSE TO REVIEWERS’ COMMENTS
Reviewer #1
1. Line 163: What is meant by "pinned" mutant Kras?
We thank Reviewer #1 for the comment. In this revised form of the paper we used the word “identified” in line 181 so that the manuscript is more comprehensible.
2. Line 232: is there a word missing after "of foreign"?
No word is missing there, but we rephrased the sentence in lines 250-251 so that it is more comprehensible.
3. Around line 249: Perhaps important to cite / mention work studying dendritic cell PDL1 in Mayoux(Chen) 2020 and Soyong(Mellman) 2020 ?
We thank Reviewer #1 for this suggestion. We have now added the proposed references in lines 268-269.
4. Line 263: Why no mention of Imfinzi or Libtayo?
We thank Reviewer #1 for the comment. We have now added the two drugs in lines 283-284.
5. Table2: The genes examined are limited by those in current common panels. Future reviews will hopefully include WES and/or a focus on IO-related genes. e.g. HLA/ Agpres.
We fully agree with Reviewer #1 that this information will greatly help to elucidate the molecular players in immune resistance phenomena in cancer.
Spella and Stathopoulos. Response to Reviewers “cancers-1039058”
6. Lines 278-284: why no mention of a role HLA by McGranahan/2017, Chowell/2019, Montesion 2020. These studies cannot be omitted.
We have now added the suggested references in line 298.
7. Point of agreement: Line 356: This reviewer fully agrees with lack of relative emphasis on frequent co-mutations vis a vis driver mutations. In fact, there is little study of "immunologic drivers".
We thank Reviewer #1 for the positive comment.

Reviewer 2 Report

Current manuscript is a very interesting literature review attempting to elucidate mechanisms of immune resistance to current therapies of Lung adenocarcinomas. 

The review is well written, though some issues require further improvement:

  1. Not all current targeted therapies and molecular targets are described in the introduction and discussion sections. 
  2. Most importantly, it seems outdated that the KRAS G12C inhibitor currently approved by FDA and under vigorous clinical investigation is not referred in the manuscript. 
  3. Exploratory analyses from the KEYNOTE-042 and KEYNOTE-189 studies presented in the ESMO immunoncology 2019 congress showed that pembrolizumab was equally effective in KRAS mutant and KRASwt patients. These data should complement preclinical data presented in this review. 

Author Response

RESPONSE TO REVIEWERS’ COMMENTS
Reviewer #2

  1. Current manuscript is a very interesting literature review attempting to elucidate mechanisms of immune resistance to current therapies of Lung adenocarcinomas.
    We thank Reviewer #2 for the positive comments.
    2. Not all current targeted therapies and molecular targets are described in the introduction and discussion sections.
    We thank Reviewer #2 for the constructive criticism. We have added more information about targeted therapies in lines 55-75 and 282-287.
    3. Most importantly, it seems outdated that the KRAS G12C inhibitor currently approved by FDA and under vigorous clinical investigation is not referred in the manuscript.
    We apologize for the neglect. We have added the relative information in lines 74-75 and 241.
    4. Exploratory analyses from the KEYNOTE-042 and KEYNOTE-189 studies presented in the ESMO immunoncology 2019 congress showed that pembrolizumab was equally effective in KRAS mutant and KRASwt patients. These data should complement preclinical data presented in this review.
    We thank Reviewer #2 for the constructive criticism. We have added this data in line 285.